# Biological Control of Root Rot of Strawberry by *Bacillus amyloliquefaciens* Strains CMS5 and CMR12

**DOI:** 10.3390/jof10060410

**Published:** 2024-06-06

**Authors:** Ruixian Yang, Ping Liu, Wenyu Ye, Yuquan Chen, Daowei Wei, Cuicui Qiao, Bingyi Zhou, Jingyao Xiao

**Affiliations:** 1School of Environmental Engineering and Chemistry, Luoyang Institute of Science and Technology, Luoyang 471002, China; lylp76lp@163.com (P.L.); luomuling520@gmail.com (Y.C.); weidaowei05@163.com (D.W.); qiaocuicui2004@outlook.com (C.Q.); zby2680641714@outlook.com (B.Z.); 13091296356@139.com (J.X.); 2China National Engineering Research Center of JUNCAO Technology, College of JunCao Science and Ecology (College of Carbon Neutrality), Fujian Agriculture & Forestry University, Fuzhou 350002, China; 3Technology Innovation Center for Monitoring and Restoration Engineering of Ecological Fragile Zone in Southeast China, Ministry of Natural Resources, Fuzhou 350002, China

**Keywords:** *Fragaria*×*ananassa*, *Fusarium solani*, *Bacillus amyloliquefaciens*, antifungal lipopeptides, biological control, growth-promoting effect

## Abstract

Strawberry root rot caused by *Fusarium solani* is one of the main diseases of strawberries and significantly impacts the yield and quality of strawberry fruit. Biological control is becoming an alternative method for the control of plant diseases to replace or decrease the application of traditional chemical fungicides. To obtain antagonistic bacteria with a high biocontrol effect on strawberry root rot, over 72 rhizosphere bacteria were isolated from the strawberry rhizosphere soil and screened for their antifungal activity against *F. solani* by dual culture assay. Among them, strains CMS5 and CMR12 showed the strongest inhibitory activity against *F. solani* (inhibition rate 57.78% and 65.93%, respectively) and exhibited broad-spectrum antifungal activity. According to the phylogenetic tree based on 16S rDNA and *gyrB* genes, CMS5 and CMR12 were identified as *Bacillus amyloliquefaciens*. Lipopeptide genes involved in surfactin, iturin, and fengycin biosynthesis were detected in the DNA genomes of CMS5 and CMR12 by PCR amplification. The genes related to the three major lipopeptide metabolites existed in the DNA genome of strains CMS5 and CMR12, and the lipopeptides could inhibit the mycelial growth of *F. solani* and resulted in distorted hyphae. The inhibitory rates of lipopeptides of CMS5 and CMR12 on the spore germination of *F. solani* were 61.00% and 42.67%, respectively. The plant-growth-promoting (PGP) traits in vitro screening showed that CMS5 and CMR12 have the ability to fix nitrogen and secreted indoleacetic acid (IAA). In the potting test, the control efficiency of CMS5, CMR12 and CMS5+CMR12 against strawberry root rot were 65.3%, 67.94% and 88.00%, respectively. Furthermore, CMS5 and CMR12 enhanced the resistance of strawberry to *F. solani* by increasing the activities of defense enzymes MDA, CAT and SOD. Moreover, CMS5 and CMR12 significantly promoted the growth of strawberry seedlings such as root length, seedling length and seedling fresh weight. This study revealed that *B. amyloliquefaciens* CMS5 and CMR12 have high potential to be used as biocontrol agents to control strawberry root rot.

## 1. Introduction

Strawberry (*Fragari×ananassa*) is the most economically important cultivated small berry plant in the world [1]. China is currently the world’s largest producer of strawberry, accounting for about one-third of the global output [2]. Nevertheless, soil-borne plant pathogens are found in most strawberry-planted soils. Among soil-borne fungal diseases, root rot is considered the most important and destructive disease, leading to serious economic losses in the strawberry industry [3]. The main symptoms of root rot disease include the gradual blackening and decay of the plant root system with consequent suppression in vigor and yield of the strawberry plant [4]. Many fungal species can cause strawberry root rot, such as *Colletotrichum* spp., *Fusarium* spp., *Verticillium* spp., *Pestalotiopsis* spp., *Neopestalotiopsis* spp., *Rhizoctonia* spp., *Phytophthora* spp., *Pythium* spp., *Ilyonectria* spp., and *Dactylonectria* spp. [5,6,7,8,9]. Among those pathogens, *Fusarium* spp. are the major agents of strawberry root rot in all the main strawberry-growing areas of the world, and *F. solani* is considered to be the major pathogen in strawberry nurseries in many parts of China, such as Liaoning province, Jiangsu province, Guizhou province, Shandong province, and Xinjiang autonomous region [10,11,12], and it has also been reported as a strawberry pathogen in Iran, Spain, Italy, and Mexico along with other pathogens [13,14,15]. The control of strawberry root rot diseases is very challenging because several fungi usually work together, giving rise to its occurrence [16]. Several approaches are currently applied to control strawberry root rot, such as the breeding and selection of resistant cultivars, crop rotation, cultivation patterns, improved drainage and chemical plant protection, individually or combined [17]. The most common approach for the prevention and control of strawberry root rot is to use chemical fungicides, but strawberry fruits are consumed directly, and fungicide applications bring a range of issues such as the accumulation of toxin residue on the fruits, increased regulation, and negative impacts on the environment and human consumers, considering the increasing demand for sustainable agriculture [18]. Therefore, it is important to find a safer and more efficient biocontrol agent to control strawberry root rot.

Biological control is an efficient and environmentally friendly way to prevent strawberry root rot. Currently, many species of antagonistic microorganisms, such as *Streptomyces* spp., *Trichoderma* spp., and *Bacillus* spp., have been widely isolated and developed as agents to prevent strawberry root rot [19,20]. *Bacillus* spp., known as plant-growth-promoting rhizobacteria (PGPR), have received considerable attention as biocontrol agents [21]. Many *Bacillus* strains have been used to promote plant growth and control soil-borne plant pathogens [22,23]. *B. amyloliquefaciens*, the type species of *Bacillus* genus, isolated from the rhizosphere, has exhibited notable antifungal properties and the ability to promote plant growth [24]. *B. amyloliquefaciens* suppress the pathogens through various mechanisms, such as the production of antibiosis, competition for places and nutrients, and induction of plant defenses [25].

Inducing disease resistance (ISR) is the one of key mechanisms of *B. amyloliquefaciens* in its biocontrol effects. *B. amyloliquefaciens* enhances plant resistance to pathogens by activating ISR system of plant, which induces the production of defensive enzymes and disrupts normal physiological metabolism of the cells, thereby enhancing the ability to resist plant pathogens [26]. Currently, the defensive enzymes related to plant resistance against pathogen invasion mainly include peroxidase (POD), superoxide dismutase (SOD), catalase (CAT), polyphenol oxidase (PPO), and phenylalanine ammonia-lyase (PAL) [27]. Additionally, some studies have shown that *B. amyloliquefaciens* can promote plant growth by secreting plant growth hormones, non-symbiotic nitrogen fixation, solubilizing insoluble soil phosphates, producing extracellular polysaccharides, volatile organic compounds, and iron carriers [28].

Although previous studies have extensively explored the use of *Bacillus* species in controlling strawberry root rot diseases, the focus has mainly been on the control the pathogens including *R. solani*, *F. oxysporum*, and *C. gloeosporioides.* Studies on the screening of *Bacillus* strains is still relatively lacking regarding strawberry root rot caused by *F. solani*. Consequently, there remains a necessity to isolate and screen biocontrol strains against the pathogen *F. solani*. To achieve this goal, the objectives of this study were as follows: (1) to isolate, screen and identify rhizobacteria for controlling strawberry root rot caused by *F. solani*; (2) to evaluate the potential of *B. amyloliquefaciens* CMS5 and CMR12 for the reduction in strawberry root rot in vitro and in vivo; (3) to explore biocontrol mechanisms of strains CMS5 and CMR12 against *F. solani*; and (4) to assess growth promotion in strawberry seedling with strains CMS5 and CMR12 through pot experiments. This study revealed that *B. amyloliquefaciens* CMS5 and CMR12 can effectively control the occurrence of strawberry root rot, and had an obvious promotion effect on strawberry seedling growth, indicating that *B. amyloliquefaciens* CMS5 and CMR12 are promising biocontrol microorganisms for controlling strawberry root rot caused by *F. solani*.

## 2. Materials and Methods

### 2.1. Plant Pathogen and Plant Materials

In this study, *Fusarium solani* was isolated from strawberry cultivars “Hongyan” by using the tissue isolation method [29]. *F. solani* was identified based on the sequence of internal transcription intervals of ribosomal DNA (which were 99.9% identical to GenBank accession number MN461539). *Neopestalotiopsis clavispora* was isolated from strawberry fruit blight in March 2022. *F. oxysporum* was isolated from tree peony root rot in September 2020. *Alternaria alternata* was isolated from peony leaf spot in May 2022. All the plant pathogens were stored in the Laboratory of Microbiology in the Luoyang Institute of Science and Technology (Luoyang, Henan province, China). One-year-old strawberry seedlings of the commercial cultivar “Hongyan” or control effects test and growth-promoting effects were provided by the “Shilixiang” strawberry seedling cultivation facility (Luoyang, Henan province, China).

### 2.2. Isolation of Bacteria from Strawberry Rhizosphere Soil

Soil samples were collected from the healthy strawberry rhizosphere soil of the “Shilixiang” strawberry planting field (112°57′14.51″ E, 34°79′42.23″ N) in Luoyang, Henan province. Five samples were taken, with approximately 1 kg soil per sample collected from the vicinity of healthy strawberry plants within a 10 cm diameter of the tap root and up to 20 cm deep. After collection, the soil from each plant was thoroughly mixed. The mixed soil was then placed in plastic bags and transported to the laboratory.

The isolation for rhizosphere strains was slightly modified from Ding et al. [30]. Briefly, the soil samples were diluted from 10^−1^- to 10^−5^-fold with sterile distilled water, and a 100 µL dilution (10^−5^-fold) was spread onto tryptic soy agar (TSA) plates and placed in 28 °C incubator for 2 days. Different single colonies were inoculated to fresh TSA. The pure strains were then stored in 20% glycerol at −80 °C.

### 2.3. Screening of Antagonistic Strains

Antimicrobial activity against plant pathogens *F. solani*, *N. clavispora*, *F. oxysporum*, and *A. alternata* was estimated by dual culture assay [31]. A fungal disc of 5 mm diameter was placed on the center of PDA plate, while the tested bacteria were parallel at 25 mm away from the center on both sides. PDA plates inoculated with plant pathogens only were used as control. The plates were kept at 28 °C for 7 days. All experiments were carried out three times with three repetitions each. The inhibition zone was recorded and the mycelial growth inhibition rate was calculated using the following formula:Inhibition rate (%) = (A − B)/A × 100
where A is the diameter of the fungal colonies grown in the control, and B is the diameter of the fungal colonies grown with the bacterial culture.

### 2.4. Identification of Strains CMS5 and CMR12

Strains CMS5 and CMR12 were cultured on NA medium (beef extract 10 g/L, peptone 5 g/L, NaCl 5 g/L, agar 15 g/L) at 28 °C for 24 h, and then identified preliminarily by Gram-staining and morphological characteristics. The Gram reaction was performed, and morphological characteristics were observed under a microscope (Olympus, Shinjuku City, Japan). Strains CMS5 and CMR12 were further identified on the basis of 16S rDNA and *gyrB* sequences analysis. Strains CMS5 and CMR12 were inoculated into NB and cultured at 28 °C with 150 rpm shaking overnight. Genomic DNA was isolated using the TIANamp Bacterial DNA Kit (Tiangen Biotech Co., Beijing, China). The extracted DNA was used as a template to amplify 16S rDNA gene and gyrase gene (*gyrB*) [32,33]. The amplified primers and PCR conditions are presented in Table 1. PCR was performed using the TIANGEN Golden Easy PCR kit (Tiangen Biotech Co., Beijing, China). The PCR products were subjected to direct automated sequencing using fluorescent terminators on an ABI 377 Prism Sequencer (Sangon Biotech Co., Shanghai, China). The sequences were confirmed with a BLAST (Basic Local Alignment Search Tool) search of the NCBI (National Center for Biotechnology Information) database (https://www.ncbi.nlm.nih.gov/, accessed on 5 March 2024), and phylogenetic trees were constructed using the neighbor-joining (NJ) method, with 1000 bootstrap replications in the MEGA 10.0 package.

### 2.5. Characterization of Antifungal Lipopeptides Substances of Strains CMS5 and CMR12

Three primer pairs were used for the detection of genes (*fenA*, *srfAA*, *ituD*) of CMS5 and CMR12 involved in lipopeptide synthesis [34,35]. The primers and PCR conditions used for amplification are presented in Table 2. PCR was performed and sequenced using the same method as in Section 2.4. The sequences were confirmed with a BLAST (Basic Local Alignment Search Tool) search of the NCBI (National Center for Biotechnology Information) database (https://www.ncbi.nlm.nih.gov/, accessed on 10 March 2024).

The production of lipopeptides of strains CMS5 and CMR12 on Landy medium was examined following a predescribed protocol of Kim et al. [36]. The antifungal activity of crude extracts of lipopeptides produced by CMS5 and CMR12 was evaluated against *F. solani* by Oxford cup method [37]. A volume of 100 μL of crude extracts was dropped into an Oxford cup placed 15 mm from the edge of a Petri plate. A plug (5 mm diameter) of *F. solani* was placed in the center. A volume of 100 μL of methanol was used as control. The plates were then incubated for 7 days at 28 °C, and the inhibition zone was recorded.

### 2.6. Effects of Lipopeptides of Strains CMS5 and CMR12 on the Pathogenic Fungal Mycelial Morphology and Spore Germination

The inhibitory effect of lipopeptide substances of strains CMS5 and CMR12 on mycelial growth was slightly modified from Pan et al. [38]. Briefly, 0.2 mL from spore suspension of *F. solani* with 1 × 10^6^ spores/mL was inoculated into 8 mL of PDB medium and incubated at 28 °C with shaking at 140 rpm for 60 h. The hyphae of *F. solani* were collected by centrifugation (10,000 rpm, 5 min) and washed with PBS buffer. Then, 0.8 mL of lipopeptides (concentration 100 mg/L) of strains CMS5 and CMR12 was added. The mixture was cultured at 28 °C with shaking at 140 rpm for 24 h. The mycelial morphology of *F. solani* was observed one time per 4 h under a microscope (Olympus). The sterile distilled water was served as a control. Each treatment was repeated three times.

For spore preparation, an equal volume of lipopeptides (concentration 100 mg/L) of strains CMS5 and CMR12 was mixed with spore suspension of *F. solani* (concentration 1 × 10^6^ spores/mL). A 10 μL drop of the mixture was placed on a concave glass slide, covered with a cover slip, and placed in a sterilized petri dish. The setup was incubated at 28 °C with moisture maintained, and spore germination was observed one time per 4 h under a microscope (Olympus). For each treatment, 100 spores were observed, and observations continued for up to 28 h. The sterile distilled water was served as a control. Each treatment was repeated three times. The germination rate and inhibition rate of pathogenic fungal spores were calculated using the following formula:Germination rate (%) = (C/D) × 100;
Inhibition rate (%) = (E − F)/E × 100.
where C is the number of germinated spores, and D is the total observed spores. E is the conidial germination rate in the control, and F is the conidial germination rate in the treatment.

### 2.7. Determination of Plant Growth-Promoting Traits of Strains CMS5 and CMR12

Strains CMS5 and CMR12 were individually inoculated on nitrogen-free Ashby medium [39] and CAS agar medium [40], and incubated at 28 °C for 3 days. The abilities to fix nitrogen and secrete iron carriers were evaluated according to the growth condition of the strains in Ashby medium and CAS agar medium [41]. For phosphate solubilization, strains CMS5 and CMR12 were inoculated on the PKO inorganic phosphate medium and incubated at 28 °C for 5 days. The phosphate solubilization capabilities were assessed by the clear zone method [42]. IAA production of strains CMS5 and CMR12 was determined using Salkowski’s reagent [43].

### 2.8. Control Effects of Strains CMS5 and CMR12 on Strawberry Root and Defensive Enzyme Activity

Strains CMS5 and CMR12 were inoculated into 5 mL of NB medium and cultured to stationary phase. Then, 100 μL of CMS5 and CMR12 NB culture was inoculated into 100 mL PDB medium and cultivated at 28 °C for 40 h at 180 rpm. The concentration of CMS5 and CMR12 PDB culture was adjusted to 1 × 10^8^ cfu/mL. *F. solani* was inoculated at the center of PDA plates at 28 °C for 7 d. Conidial suspensions (1 × 10^8^ spores/mL) of pathogen were prepared and then stored at 4 °C for later use. One-year-old strawberry seedlings of the commercial cultivar “Hongyan” were used. The experiment included five treatments: (1) strawberry seedling+water (T1); (2) strawberry seedling+*F. solani* (T2); (3) strawberry seedling+*F. solani+*CMS5 (T3); (4) strawberry seedling+*F. solani+*CMR12 (T4); and (5) strawberry seedling+*F. solani+*CMS5+CMR12 (T5). After strawberry seedling acclimation for 20 d, 5 mL *F. solani* conidia (1 × 10^8^ spores/mL) was inoculated with strawberry roots. After 5 days of *F. solani* treatment, 20 mL CMS5 and CMR12 PDB culture were used to inoculate the soil around each seedling through a root irrigation method. Each treatment included 5 pots with three repetitions each. All the treatments were incubated for 60 days at 25 °C and 80% relative humidity. The disease severity of the strawberry seedlings was assessed using a scoring system of 0–5, from the report of Vestberg et al. [44]. The germination rate and inhibition rate of pathogenic fungal spores were calculated using the following formula:Disease Index = ∑ (disease level × number of plants at that level)/(total number of plants × highest disease level) × 100;
Control efficiency (%) = (control disease index − treatment disease index)/control disease index × 100.

To determine the induced resistance to *F. solani* by strains CMS5 and CMR12, strawberry seedlings leaves were randomly collected at 0, 36, 72, 120, and 168 h after five treatments (T1, T2, T3, T4 and T5) to detect the activity of superoxide dismutase (SOD), malondialdehyde (MDA) and catalase (CAT). MDA content in strawberry leaves was determined using the thiobarbituric acid method [45]. CAT activity was measured using ultraviolet spectrophotometry, and SOD activity was determined using the nitroblue tetrazolium method [46]. All experiments were conducted in triplicate.

### 2.9. Growth-Promoting Effects of Strains CMS5 and CMR12 on Strawberry Seedlings

The same method used in Section 2.8 was used in this experiment, which consisted of four treatments: (1) strawberry seedling+water (T1); (2) strawberry seedling+CMS5 (T2); (3) strawberry seedling+CMR12 (T3); and (4) strawberry seedling+CMS5+CMR12 (T4). Each treatment included 5 pots with three repetitions each. All the treatments were incubated 60 days at 25 °C and 80% relative humidity. The strawberry seedlings were carefully excavated, and their height, root length, and fresh weight were measured.

### 2.10. Data Statistics and Analysis

Data obtained from the experiments were processed using a one-way analysis of variance (ANOVA) was performed using DPS 7.05 statistical software. Duncan’s new multiple range test was used to assess the significant differences, and the significance level was set at *p* < 0.05.

## 3. Results

### 3.1. Isolation and Screening of Rhizosphere Soil Bacteria

The total 72 bacterial strains were isolated from strawberry rhizosphere soil. Their antagonistic activity against *Fusarium solani* was measured using plate confrontation method. The results showed that two strains CMS5 and CMR12 exhibited strong antagonistic activity against *F. solani*. The inhibition zones of CMS5 and CMR12 against *F. solani* were 5.00 ± 0.00 mm and 6.00 ± 0.10 mm, respectively. The inhibition rates of CMS5 and CMR12 against *F. solani* were 57.78% and 65.93%, respectively (Table 3, Figure 1). Meanwhile, CMS5 and CMR12 also exhibited inhibitory effects against *F. oxysporum*, *Neopestalotiopsis clavispora*, and *Alternaria alternata*. The strain CMS5 displayed inhibition zones of 3.70 mm, 2.30 mm, and 10.10 mm against these pathogens, and the inhibition rates to the three pathogens were 65.92%, 59.26%, and 70.37%, respectively. The strain CMR12 had inhibition zones of 4.30 mm, 4.00 mm, and 11.30 mm against the three pathogens, with inhibition rates of 70.37%, 61.48%, and 71.85%, respectively(Table 3, Figure 1).

### 3.2. Identification of Strains CMS5 and CMR12

The strains CMS5 and CMR12 formed circular, milky white colonies with neat edges, a moist surface, slight convexity, and opacity on NA medium. CMS5 and CMR12 were Gram-positive and rod-shaped, rounded at both ends. Based on 16S rDNA sequence alignment, strains CMS5 and CMR12 obtained 1450 bp and 1443 bp sequence fragments, with GenBank accession numbers PP338070 and PP338071, respectively. The *gyrB* genes sequence fragments of strains CMS5 and CMR12 were 1162 bp and 1152 bp, and the GenBank accession numbers were PP355738 and PP355739, respectively. Phylogenetic trees constructed based on the 16S rDNA and *gyrB* gene sequences showed that strains CMS5, CMR12 and *Bacillus amyloliquefaciens* were in the same branch (Figure 2A,B). Based on the combined morphological and molecular characteristics, strains CMS5 and CMR12 were identified as *B. amyloliquefaciens*.

### 3.3. Antifungal Activity of Lipopeptide Substances of Strains CMS5 and CMR12

Gene fragments of *ituD*, *fenA*, and *srfAA* were obtained from the DNA of strains CMS5 and CMR12. To further confirm the genes corresponding to the antimicrobial substances, gene-specific bands were recovered and cloned for sequencing, and sequence analysis was performed using BLASTX (https://www.ncbi.nlm.nih.gov/blast.cgi, accessed on 10 March 2024) in GenBank. It was found that the sequences of the *ituD*, *fenA*, and *srfAA* gene fragments from strains CMS5 and CMR12 exhibited 99% similarity to the sequences of the non-ribosomal peptide synthetase proteins from *B. amyloliquefaciens*, specifically the iturin A, bacillomycin D synthetase, fengycin synthetase, and surfactin synthetase proteins, respectively (Table 4). The results indicated that the genes related to the three major metabolites (fengycin, surfactin and iturin) exist in the genomes of strains CMS5 and CMR12. The lipopeptide extract of strains CMS5 and CMR12 had an obvious inhibitory effect on *F. solani* in vitro. The diameters of inhibition zones induced by lipopeptide extract were 7.50 ± 0.20 mm and 10.50 ± 0.45 mm, respectively (Figure 3).

### 3.4. Inhibitory Effects of Lipopeptides of Strains CMS5 and CMR12 on the Pathogenic Fungal Mycelial Morphology and Spore Germination

As shown in Figure 4, 12 h after treatment with the lipopeptides of strains CMS5 and CMR12, the mycelia of *F. solani* exhibited partial staining. Some hyphal tips were swollen, and there was an increase in hyphal septum thickness. After 16 h, nearly all the mycelia of *F. solani* were stained, with a large number of hyphal tips swollen and a few hyphae breaking. After 20 h, the permeability of the treated hyphae further increased, extensive staining of the hyphal was observed, and numerous hyphae were broken. After 24 h, the mycelia of *F. solani* treated with the lipopeptides fragmented completely. In contrast, the mycelial morphology of *F. solani* treated with distilled water remained smooth and intact, and produced conidia. The results indicate that the lipopeptides of strains CMS5 and CMR12 can cause deformities and breakage of the mycelia of of *F. solani*, increasing the permeability of hyphal cells’ membranes.

To further understand the effects of lipopeptides of CMS5 and CMR12 on spores of *F. solani*, the germination and inhibition rates of spores were analyzed. As shown in Figure 5A,B, the spore germination of *F. solani* was inhibited. After 8 h treatment with lipopeptides of CMS5 and CMR12, the spore germination rates were only 9.00% and 4.00%, and the spore germination rate of the control treatment was 39.00%. The inhibition rates of lipopeptides of CMS5 and CMR12 were 76.76% and 90.15%, respectively. After 28 h, all spores of the control group had germinated, whereas the germination rates of lipopeptides of CMS5 and CMR12 were 61.00% and 42.67%, with inhibition rates of 39.00% and 57.00%. Microscopic observation showed that spores treated with lipopeptides were deformed, swelling at the ends or middle. Conversely, the spores germinated normally and formed mycelia in the control. The results indicated that the lipopeptides of CMS5 and CMR12 significantly delay the germination of spores of *F. solani.* Additionally, the lipopeptides of CMS5 and CMR12 also caused spores deformities, thereby inhibiting their germination.

### 3.5. Plant Growth-Promoting Traits of Strains CMS5 and CMR12

Strains CMS5 and CMR12 successfully grew on Ashby medium, indicating that they had nitrogen-fixing capabilities (Figure 6A). However, strains CMS5 and CMR12 could not dissolve phosphate and did not produce siderophore. The supernatant of strains CMS5 and CMR12 was mixed with the Salkowski reagent, and the mixed solution turned pink, indicating that CMS5 and CMR12 produced IAA. The concentration of IAA synthesized by strains CMS5 and CMR12 was 25.68 mg/L and 32.97 mg/L (Figure 6B).

### 3.6. Biocontrol Efficiency of Strains CMS5 and CMR12 against Strawberry Root Rot

The incidence of strawberry root rot on each treatment was investigated after inoculation for 60 days (Table 5). The results showed that strains CMS5 and CMR12 had a strong control effect on root rot of strawberry (Figure 7). As shown in Table 5, the disease index of the control group (T2) inoculated only with the pathogen was 83.30 ± 1.66, whereas the disease index of the CMS5-treated group (T3), CMR12-treated group (T4), and CMS5+CMR12-treated group (T5) were 28.90 ± 1.57, 26.70 ± 0.80, and 10.00 ± 1.20, respectively. The biocontrol efficiency of T3, T4 and T5 group against strawberry root rot were 65.3 ± 0.07%, 67.94 ± 0.10% and 88.00 ± 0.06%, respectively. The strawberry treated with the distilled water group showed good growth and no signs of disease. The results indicated that strains CMS5 and CMR12 effectively controlled the occurrence of potted strawberry root rot, and the CMS5+CMR12-treated group showed significantly better biocontrol efficiency than the individual strain treatments.

### 3.7. Effects of Strains CMS5 and CMR12 on the Activity of Strawberry Defense-Related Enzyme Activities

To determine whether the defense system is activated in response to strains CMS5, CMR12 against *F. solani*, the activity of three defense enzymes (MDA, CAT, and SOD) were selected. Compared with the T2 group (inoculated only with the pathogen), the content of MDA in the T3, T4 and T5 groups (CMS5-treated group, CMR12-treated group, and CMS5+CMR12-treated group) began to decrease 36 h after inoculation. The MDA content was lower in the T5 group than in the T3 and T4 groups, indicating that CMS5 and CMR12 effectively inhibit MDA synthesis in strawberry leaves, with CMS5+CMR12-treated group proving more effective than treatments with individual strains (Figure 8A). The activities of CAT and SOD in the T2 group began to decrease 36 h after inoculation, while the activities of CAT and SOD were significantly higher in the T3, T4 and T5 groups. Seventy-two hours after inoculation, the activities of CAT and SOD in the T5 group were higher than in the T2, T3 and T4 groups (Figure 8B,C). These results showed that strains CMS5 and CMR12 can induce an increase in the defensive enzyme activities of CAT and SOD in strawberries, and CMS5+CMR12-treated group proved more effective than treatments with individual strains. Therefore, strains CMS5 and CMR12 could rapidly activate the defense enzyme system in strawberry plants, enabling strawberries to respond rapidly to *F. solani* infection and improve their disease resistance.

### 3.8. Growth Promotion Effects of Strains CMS5 and CMR12 on Strawberry Seedlings

In the greenhouse, after the strawberry seedlings were treated with strains CMS5 and CMR12 fermentation broth for 60 days, compared with the control, strains CMS5 and CMR12 significantly enhanced the plant height, root length, and fresh weight of the strawberry seedlings (Table 6, Figure 9). The growth promotion rates for strain CMS5 on plant height, root length, and fresh weight were 8.89%, 34.18%, and 99.84%, respectively. For strain CMR12, the corresponding rates were 42.22%, 18.21%, and 95.62%, respectively. When strains CMS5 and CMR12 were co-inoculated, the growth promotion rates for plant height, root length, and fresh weight reached 25.65%, 26.51%, and 159.16%, respectively.

## 4. Discussion

### 4.1. Significance of Exploring Biocontrol Resources for Plant Disease

Strawberry root rot is a complex disease that significantly challenges agricultural production, especially in protected cultivation environments where economic losses can be substantial [47]. *Bacillus amyloliquefaciens* is known for producing various secondary metabolites and exhibits broad-spectrum antifungal activities. It is widely used as a biocontrol agent in the management of crop diseases and also acts as a plant-growth-promoting rhizobacterium, significantly enhancing plant growth [48]. Some studies have demonstrated that *B. amyloliquefaciens* play a pivotal role in the control of strawberry root rot. Wu et al. [49] found that strains *B. amyloliquefaciens* PMB04 and PMB05 have demonstrated effective antagonism against *Colletotrichum gloeosporioides*, significantly reducing the incidence of anthracnose on strawberry seedlings. Chen et al. [50] obtained a strain of *B. amyloliquefaciens* CM3, which achieved a control efficacy of 64.86% against root rot caused by *F. oxysporum*, and promoted strawberry growth, indicating that strain CM3 is suitable for development into a microbial formulation. These studies provide essential microbial resources for the biological control of strawberry root rot. However, due to the diverse composition of root rot pathogens, most studies have focused on *B. amyloliquefaciens* against *C. gloeosporioides* and *F. oxysporum* [51,52]. Studies focusing on *F. solani*, a pathogen associated with strawberry root rot, are scarce. This study focuses on *F. solani* as the target pathogen, obtaining two strains of *B. amyloliquefaciens*, CMS5 and CMR12, which have demonstrated significant biocontrol efficacy against strawberry root rot and notably promote the growth of strawberry seedlings. These findings contributed to the development of biocontrol resources for managing strawberry root rot and broadened the potential applications of *B. amyloliquefaciens* in agriculture.

A potential biocontrol strain may have a broad antimicrobial spectrum. Previous studies have shown that most antagonistic *B. amyloliquefaciens* strains have broad-spectrum activity [53]. The type strain of the *Bacillus* species, *B. amyloliquefaciens* FZB42, is commercially used as biocontrol bacterium, being especially efficient against a broad spectrum of pathogens, such as *F. graminearum*, *Rhizoctonia solani*, *Magnaporthe oryzae*, and *Phytophthora nicotianae* [54]. Similarly, in present study, we screened two broad-spectrum bacteria, CMS5 and CMR12, which exhibited obvious inhibitory activity against plant pathogenic *Alternaria alternata*, *F. oxysporum*, and *Neopestalotiopsis clavispora* on PDA medium. Strains CMS5 and CMR12 showed an especially strong effect in inhibiting the mycelial growth of *A. alternata*. The results showed that strains CMS5 and CMR12 may have potential application prospects in controlling other fungal diseases.

### 4.2. Biocontrol Mechanisms of B. amyloliquefaciens CMS5 and CMR12

Current research indicates that the suppression of various plant diseases by *B. amyloliquefaciens* involves mechanisms including competitive action, antibiotics, inducing plant resistance (ISR), and plant-growth promotion. Antibiotics are a pivotal mechanism in controlling plant diseases through the secretion of lipopeptide substance, polyketide compounds, and antimicrobial proteins [55]. Among these secondary metabolites, lipopeptides are extensively studied as antimicrobial agents and are categorized into three main groups based on their structure: iturins, surfactins, and fengycins [56]. Studies have demonstrated that the inhibitory effects of lipopeptides on pathogens primarily manifest in the suppression of mycelial growth and impact spore germination. For instance, Kang et al. [57] reported that the culture filtrate of *B. amyloliquefaciens* PPL containing fengycin can inhibit plant wilt diseases caused by *F. oxysporum* both in vitro and in vivo by affecting the growth of spores and mycelia. Jiao et al. [58] discovered that bacillomycin D produced by *B. amyloliquefaciens* YN201732 could inhibit spore germination of powdery mildew, effectively controlling tobacco powdery mildew. Additionally, Chowdhury et al. [59] found that the bacillomycin D of *B. amyloliquefaciens* FZB42 could directly inhibit the growth of *R. solani* in lettuce and alter mycelial morphology, conidial cell walls, and cytoplasmic membranes. In this study, lipopeptides of strains CMS5 and CMR12 exhibited strong inhibitory effects on *F. solani* causing strawberry root rot. The biocontrol mechanisms primarily involve causing deformities and breakage of *F. solani* mycelia, delaying spore germination, and inducing deformities in the spores. These results aligned with previously reported studies on the biocontrol mechanisms of *B. amyloliquefaciens*. Future work could aim to further purify and identify the lipopeptide substances produced by strains CMS5 and CMR12 that are crucial for antimicrobial activity. Techniques such as scanning electron microscopy, fluorescence microscopy, and comparative transcriptomic analysis could elucidate the antimicrobial mechanisms of high-purity lipopeptide metabolites from CMS5 and CMR12 against *F. solani* at cellular, physiological, biochemical, and molecular biology levels.

ISR is another key mechanism of *B. amyloliquefaciens* in its biocontrol effects. ISR is induced by non-pathogenic strains, attenuated strains, proteins or glycoproteins, extracellular polysaccharides, and lipopolysaccharides, which activate defense enzymes in the plant such as PAL, POD, SOD, CAT, and PPO. These enzymes facilitate the development of ISR, enhancing the ability of plants to resist pathogen invasion [60,61]. SOD and CAT are integral to the intracellular defense enzyme system; the accumulation of reactive oxygen species and free radicals can damage the cell membrane system, affecting plant health. The function of SOD and CAT is to scavenge reactive oxygen species and free radicals to enhance plant resistance to pathogens [62,63]. Changes in the content of MDA are directly related to plant disease resistance. When plants are damaged, the disruption of the reactive oxygen metabolism system leads to an increase in the MDA content in lipids, increases membrane permeability, and reduces plant resistance [64,65]. Pei et al. [66] showed that *B. amyloliquefaciens* Oj-2.16 enhances resistance in tomato plants to *Verticillium* wilt by increasing the activities of defense-related enzymes CAT and SOD. Compared to tomato seedlings inoculated with *V. dahliae*, the treated with strain Oj-2.16 exhibited significantly reduced MDA content. Our study found that after 36 **h** of treatment with strains CMS5 and CMR12, there was an increasing trend of the activities of CAT and SOD, and a decreasing trend of MDA content, whereas in the control group inoculated only with the pathogen, the activities of CAT and SOD decreased, and MDA content increased. This result showed that strains CMS5 and CMR12 increased strawberry resistance to root rot by increasing the activity of some defense. However, the mechanisms of strains CMS5 and CMR12 inducing systemic resistance in strawberries remain to be fully elucidated. Previous studies have suggested that *B. amyloliquefaciens* may contribute to inducing plant systemic resistance through the production of various secondary metabolites. Therefore, there is a need to further explore the effects of the secondary metabolites of CMS5 and CMR12 on inducing systemic resistance in strawberries through gene expression analysis, proteomics and metabolomics analysis.

### 4.3. Practical Applications of B. amyloliquefaciens CMS5 and CMR12

This study demonstrated that *B. amyloliquefaciens* CMS5 and CMR12 exhibit antimicrobial effects and induce systemic resistance against *F. solani*, which causes strawberry root rot. Under greenhouse conditions, strains CMS5 and CMR12 effectively control the occurrence of strawberry root rot. However, the biocontrol mechanisms at various interaction stages between the plant and the pathogen still require further investigation to provide a theoretical basis for the application of *B. amyloliquefaciens* CMS5 and CMR12 under strawberry field conditions. Consequently, further exploration of interactions among *B. amyloliquefaciens*, the root rot pathogen, and the host will be a primary focus of future research. Additionally, investigating the synergistic effects of CMS5 and CMR12 with other strains, chemical pesticides and fertilizers, and assessing their effectiveness under field conditions to develop integrated antimicrobial and fertilizing solutions, will also be a key point of future work.

## 5. Conclusions

In summary, *B. amyloliquefaciens* CMS5 and CMR12 were obtained among 72 rhizosphere bacteria strains as potent biocontrol agent against *F. solani*, the pathogenic fungi causing strawberry root rot. The results of the pot experiment demonstrated that *B. amyloliquefaciens* CMS5 and CMR12 effectively inhibited root rot and significantly enhanced the growth of strawberry seedlings. These findings indicate that *B. amyloliquefaciens* CMS5 and CMR12 have great potential as a biocontrol resource for preventing and controlling strawberry root rot, making it a promising candidate for future development.

## Figures and Tables

**Figure 1 jof-10-00410-f001:**
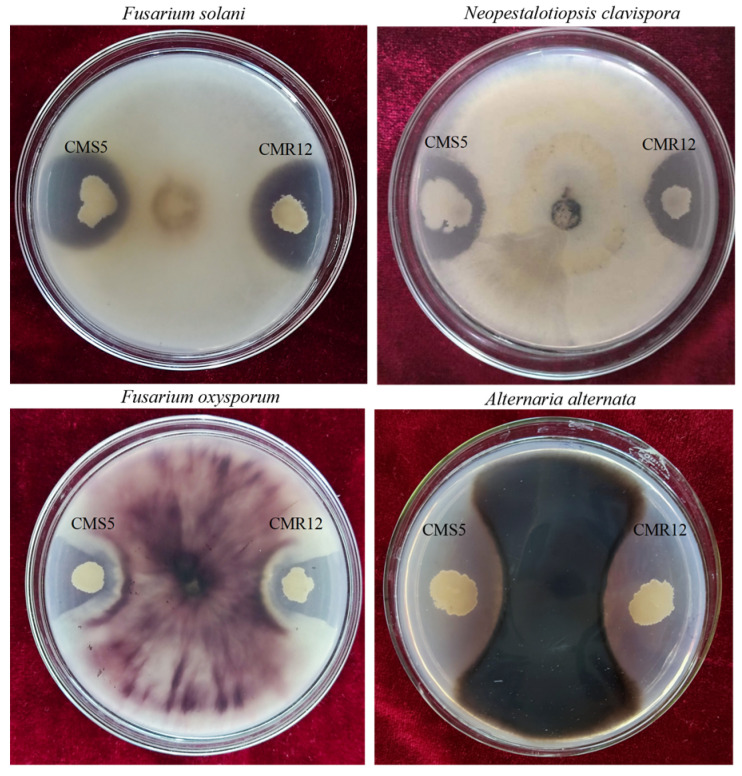
Inhibitory effect of the rhizosphere bacteria against four different plant pathogens on PDA plates.

**Figure 2 jof-10-00410-f002:**
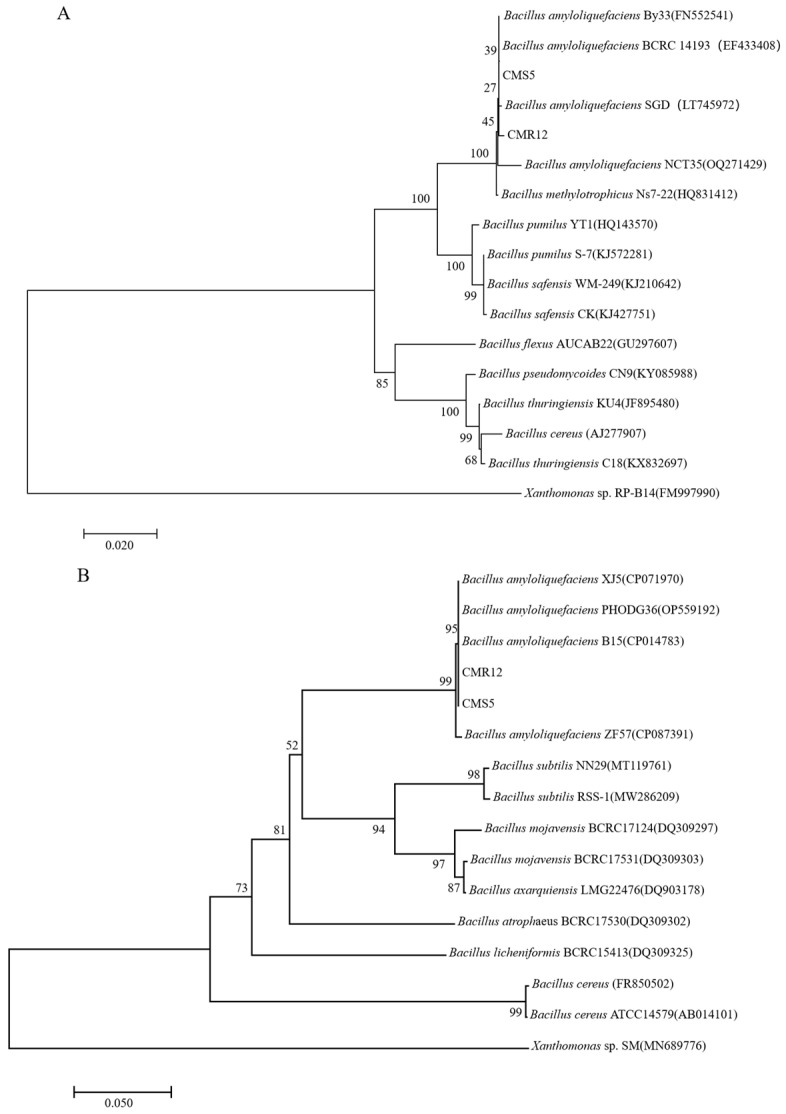
Phylogenetic trees based on the 16S rDNA sequence (**A**) and *gyrB* sequence (**B**) of CMS5 and CMR12 and their homologous sequences. Phylogenetic trees were constructed by the neighbor-joining method of MEGA10.0 with bootstrap values based on 1000 replications. *Xanthomonas* sp. RP-B14 and *Xanthomonas* sp. SM were chosen as the outgroup. Gene accession numbers of bacterial strains are indicated in parentheses. The scale bar represents the number of substitutions per base position.

**Figure 3 jof-10-00410-f003:**
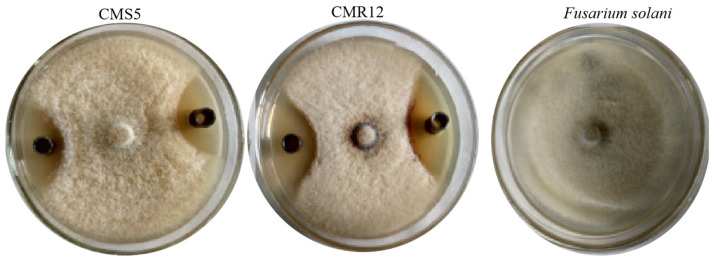
Inhibition effect of lipopeptides of strains CMS5 and CMR12 against *F. solani* on PDA plates.

**Figure 4 jof-10-00410-f004:**
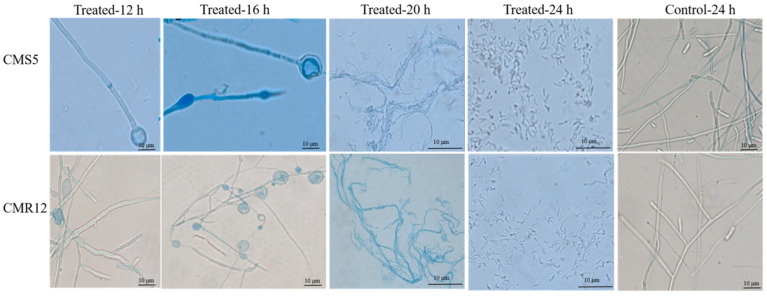
Inhibitory effects of lipopeptide substances of strains CMS5 and CMR12 on the mycelial morphology of *F. solani*.

**Figure 5 jof-10-00410-f005:**
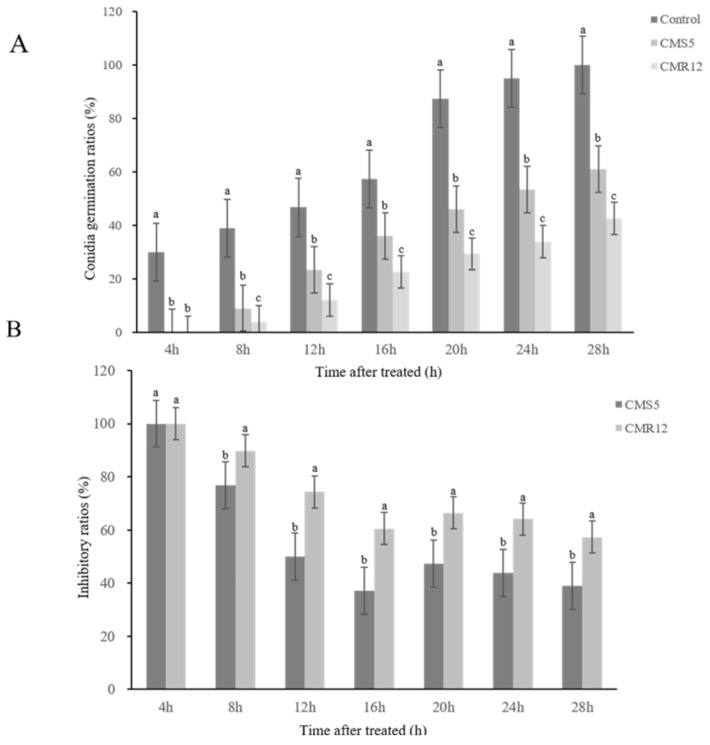
Inhibitory effects of lipopeptides of strains CMS5 and CMR12 on the conidia germination of *F. solani*. (**A**) Conidia germination ratios; (**B**) inhibitory ratios of conidia germination. Bars indicate the standard error of the mean. Columns marked with the same letter are not significantly different according to Duncan’s multiple range test at *p* < 0.05.

**Figure 6 jof-10-00410-f006:**
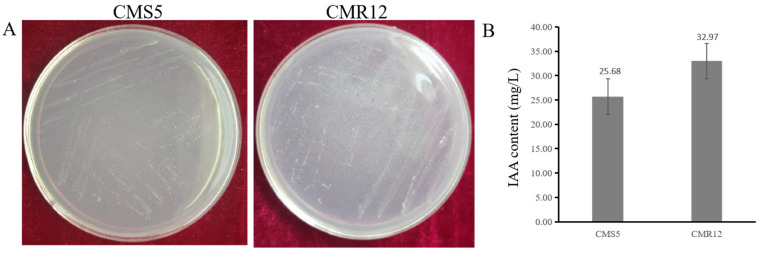
Evaluation of growth-promoting traits of strains CMS5 and CMR12. (**A**) Nitrogen-fixing capabilities of CMS5 and CMR12 on Ashby’s medium; (**B**) the concentration of IAA synthesized by strains CMS5 and CMR12 by Salkowski colorimetric method.

**Figure 7 jof-10-00410-f007:**
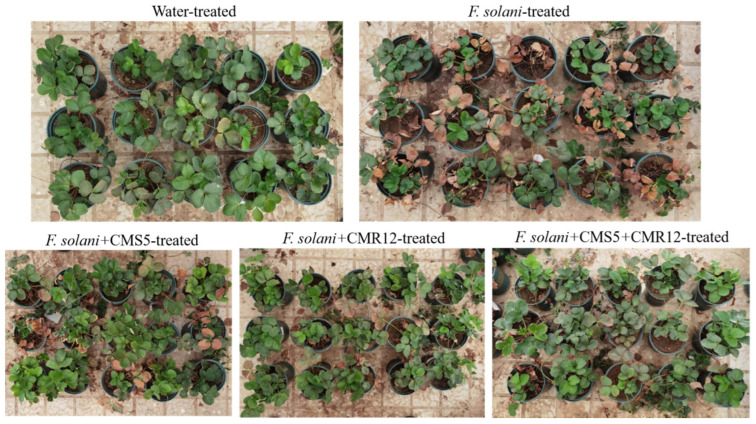
Biocontrol efficiency of CMS5 and CMR12 against strawberry root rot by different treatments.

**Figure 8 jof-10-00410-f008:**
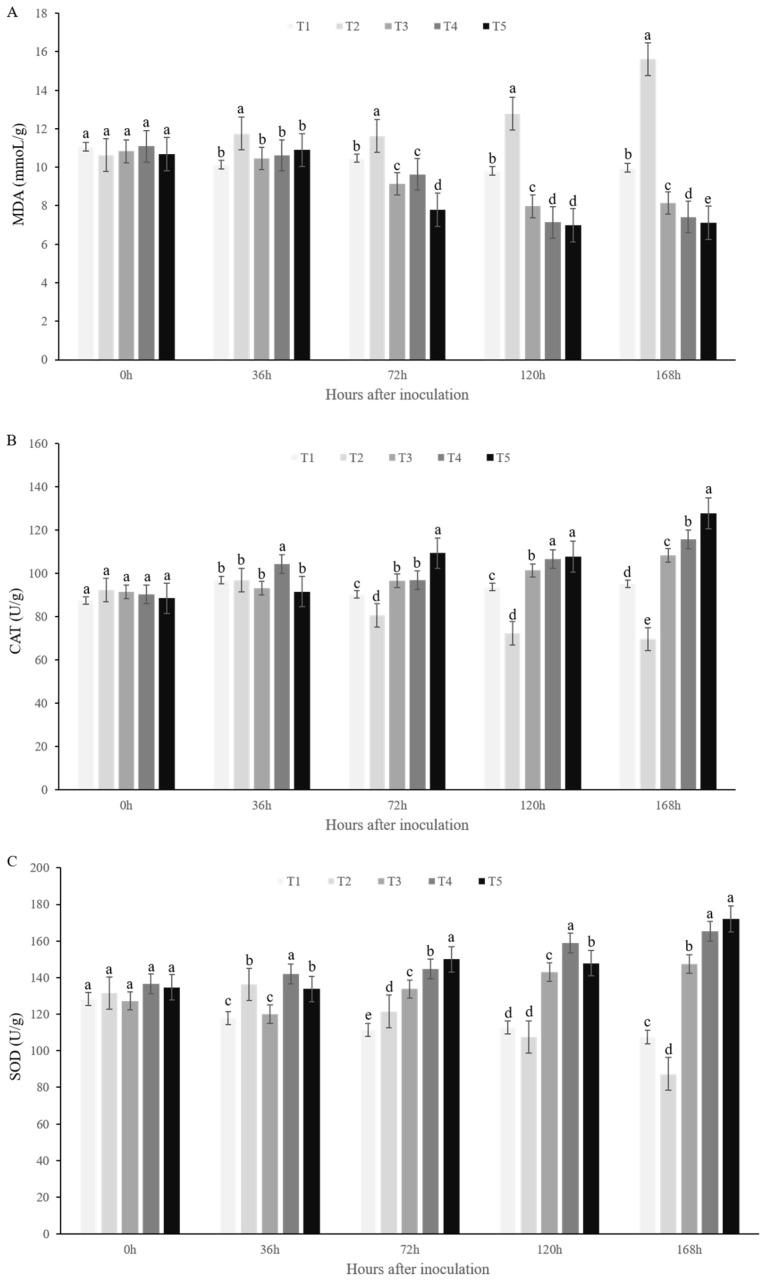
Effect of CMS5 and CMR12 on defense enzyme activity in strawberry leaves inoculated with pathogen *F. solani*. (**A**) MDA activity; (**B**) CAT activity; (**C**) SOD activity. T1, water-treated; T2, *F. solani*-treated; T3, *F. solani*+CMS5-treated; T4, *F. solani*+CMR12-treated; T5, *F. solani*+CMS5+ CMR12-treated. Bars indicate the standard error of the mean. Columns marked with the same letter are not significantly different according to Duncan’s multiple range test at *p* < 0.05.

**Figure 9 jof-10-00410-f009:**
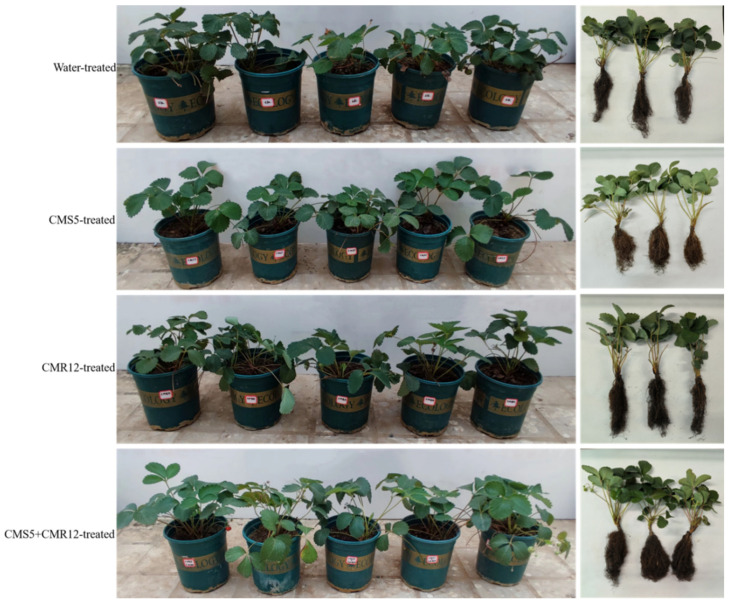
Growth promoting effects of strains CMS5 and CMR12 on strawberry seedings.

**Table 1 jof-10-00410-t001:** Amplification sites, primer sequences and PCR conditions of antagonistic bacteria.

Gene	Primer	Primer Sequence (5′-3′)	PCR Conditions	Reference
16S rRNA	27F	AGAGTTTGATCCTGGCTCAG	94 °C for 5 min (94 °C for 30 s, 55 °C for 30 s, and 72 °C for 1 min) × 35 cycles, 72 °C for 10 min	[32]
	1492R	TACGGCTACCTTGTTACGACTT		
*gyrB*	UP-1S	GAAGTCATCATGACCGTTCTGCA		[33]
	UP-2Sr	AGCAGGGTACGGATGTGCGAGCC		

**Table 2 jof-10-00410-t002:** Primer sequence and PCR conditions for lipopeptide biosynthesis genes of antagonistic bacteria CMS5 and CMR12.

Gene	Primer	Primer Sequence (5′-3′)	Lipopeptide	PCR Conditions	Reference
*fenA*	FenAa	AAGAGATTCAGTAAGTGGCCCATCCAG	Fengycin	94 °C for 5 min (94 °C for 30 s, 55 °C for 30 s, and 72 °C for 1 min) × 35 cycles, 72 °C for 10 min	[34]
	FenAb	CGCCCTTTGGGAAGAGGTGC			
*srfAA*	Srfkn-1	AGCCGTCCTGTCTGACGACG	Surfactin		[35]
	Srfkn-2	TCTGCTGCCATACCGCATAGTC			
*ituD*	ItuD-f	ATGAACAATCTTGCCTTTTTA	Iturin		
	ItuD-r	TTATTTTAAATTCCCCAATT			

**Table 3 jof-10-00410-t003:** Inhibitory effect of the rhizosphere bacteria CMS5 and CMR12 against four different plant pathogens on PDA plates.

Pathogenic Strains	CMS5		CMR12	
	Inhibition Zones (mm)	Inhibitory Rate (%)	Inhibition Zones (mm)	Inhibitory Rate (%)
*F. solani*	5.00 ± 0.00 a	57.78 ± 0.04 a	6.00 ± 0.10 a	65.93 ± 0.01 a
*F. oxysporum*	3.70 ± 0.12 a	65.92 ± 0.01 b	4.30 ± 0.06 a	70.37 ± 0.03 a
*N. clavispora*	2.23 ± 0.06 b	59.26 ± 0.01 b	4.00 ± 0.10 a	61.48 ± 0.01 a
*A. alternata*	10.01 ± 0.12 a	70.37 ± 0.03 a	11.3 ± 0.06 a	71.85 ± 0.01 a

Note: Data were mean ± SD of triplicates. Means were tested with Duncan’s multiple range test for the last column. Means followed by the same letter are not significantly different (*p* < 0.05) for the same column. Lowercase letters in the same column indicate differences between CMS5 and CMR12 against for each pathogenic strain.

**Table 4 jof-10-00410-t004:** Mimilar protein sequences of lipopeptide synthetase genes of strains CMS5 and CMR12 generated from NCBI using BLASTX.

Strain	Genes	Product Size (bp)	Similar Protein Sequences	Accession Numbers	Identity (%)
CMS5	*ituD*	1122	bacillomycin D biosynthesis malonyl-CoA transacylase BamD (*B. amyloliquefaciens*)	WP_061860597	99%
	*fenA*	1365	non-ribosomal peptide synthetase (*B. amyloliquefaciens*)	WP_060674287	99%
	*srfAA*	1464	surfactin non-ribosomal peptide synthetase SrfAA (*B. amyloliquefaciens*)	WP_223255265	99%
CMR12	*ituD*	1121	bacillomycin D biosynthesis malonyl-CoA transacylase BamD (*B. amyloliquefaciens*)	WP_094246888	99%
	*fenA*	1369	non-ribosomal peptide synthetase (*B. amyloliquefaciens*)	WP_243939152	99%
	*srfAA*	1463	surfactin non-ribosomal peptide synthetase SrfAA (*B. amyloliquefaciens*)	WP_241457877	99%

**Table 5 jof-10-00410-t005:** Biocontrol efficiency of CMS5 and CMR12 against strawberry root rot by different treatments.

Treatments	Disease Index	Biocontrol Efficiency (%)
T1 (Water-treated)	0.00 ± 0.00 d	-
T2 (*F. solani*-treated)	83.30 ± 1.66 a	-
T3 (*F. solani+*CMS5-treated)	28.90 ± 1.57 b	65.30 ± 0.07 b
T4 (*F. solani+*CMR12-treated)	26.70 ± 0.80 b	67.94 ± 0.10 b
T5 (*F. solani+*CMS5*+*CMR12-treated)	10.00 ± 1.20 c	88.00 ± 0.06 a

Note: Data are mean ± SD of triplicates. Means were tested with Duncan’s multiple range test for the last column. Means followed by the same letter are not significantly different (*p* < 0.05) for the same column.

**Table 6 jof-10-00410-t006:** Growth--promoting effects of CMS5 and CMR12 on strawberry seedings.

Treatments	Plant Height (cm)	Root Length (cm)	Total Fresh Weight (g)
T1 (water)	20.86 ± 0.65 b	16.50 ± 0.36 c	14.57 ± 1.15 c
T2 (CMS5)	28.00 ± 0.78 a	17.97 ± 0.76 c	29.12 ± 2.83 b
T3 (CMR12)	24.67 ± 2.32 a	23.47 ± 0.83 a	28.50 ± 1.69 b
T4 (CMS5+CMR12)	26.40 ± 2.94 a	20.73 ± 4.84 ab	37.76 ± 4.78 a

Note: Data are mean ± SD of triplicates. Means were tested with Duncan’s multiple range test for the last column. Means followed by the same letter are not significantly different (*p* < 0.05) for the same column.

## Data Availability

The original contributions presented in the study are included in the article, further inquiries can be directed to the corresponding author.

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
