# Peer review of "Biological Control of Root Rot of Strawberry by Bacillus amyloliquefaciens Strains CMS5 and CMR12"

_jof, 2024, doi:10.3390/jof10060410_

Round 1

Reviewer 1 Report

The topic of the work is very interesting, although there are a numerous works about the effect of biocontrol agents on pathogens, the study of native strains is relevant. 

I consider that not all tables and figures are necessary. For example 
figure 4 is not very clear, the same with figure 5 C.

Author Response

Dear Reviewer:

Thank you very much for your comments and professional advice.These opinions help to improve academic rigor of our article. Based on your suggestion  not all tables and figures are necessary, for example figure 4 and figure 5 C, we have made revised modifications on the revised manuscript with red.

Answer:

  • We have deleted figure 5 Cin the revised paper.
  • We have retained the Figure 4. Because the Figure 4 represented the results of the inhibitory effects of lipopeptide substancesof strains CMS5 and CMR12 on the mycelial morphology of solani. In order to better emerge the changes of mycelial morphology of F. solani in different treatment times, we used different magnifications. After 12 and 16 hours treated with lipopeptide substances, we used 40×20 magnification. After 20 and 24 hours, we used 20×20 magnification to better observe the breakage of numerous hyphae.

Reviewer 2 Report

Dear Editor and Authors,

Thank you for giving me the opportunity to review your interesting manuscript. Overall, I found the study to be informative and it is a very relevant topic despite the typos in the text.However, I have a few comments and suggestions that I believe will further improve the clarity and comprehensiveness of the manuscript.

Line 23: "amyloliquefaciens.Lipopeptide" should be "amyloliquefaciens. Lipopeptide" (missing space).

Line 24: "CMS5 and CMR12 DNA genome" should be "the DNA genomes of CMS5 and CMR12."

Line 28: "showed that CMS5 and CMR12 with the ability of fix nitrogen" should be "showed that CMS5 and CMR12 have the ability to fix nitrogen".

Line 33: "seedling dry weight.This study revealed" should be "seedling dry weight. This study revealed" (missing space).

108 - A. alternata

129-130 "hyphal" should be " deposited "

140-141 The Gram reaction was performed, and morphological characteristics were observed under a microscope (Olympus).

161 –Consider rephrasing to «The primers and PCR conditions used for amplification are presented in Table 2.»

Line 187: "hyphal" should be "hyphae", "1,0000 rpm" should be "10,000 rpm".

Tab. 3 – A. alternata

Fig. 1 It's intriguing that the A. alternata strain is the most suppressed! It should be discussed

Line 312: The phrase "combined morphological characteristics and molecular biology" should be revised to improve clarity. It should be "combined morphological and molecular data".

or:

Based on the combined morphological and molecular characteristics, strains CMS5 and CMR12 were identified as B. amyloliquefaciens.

Line 316: The phrase "in the DNA genome" is redundant; "DNA" or "genome" alone is sufficient.

Corrected text:

The gene fragments of ituD, fenA, and srfAA were obtained from the DNA of strains CMS5 and CMR12.

Line 350: "Treated" should be "treatment" to maintain correct grammar.

Twelve hours after treatment with the lipopeptides of strains CMS5 and CMR12, the

Line 351: Repetition of "of".

Corrected text:

Twelve hours after treatment with the lipopeptides of strains CMS5 and CMR12, the mycelia of F. solani exhibited partial staining.

Line 363: "with" two times
Line 425: "on the 36 h" should be "36 hours".

36 hours after inoculation.

Line 429: "on the 36 h" should be "36 hours".

36 hours after inoculation,

Line 430: "the activities of CAT and SOD were" should be "while the activities of CAT and SOD were".

while the activities of CAT and SOD were

Line 431: "on the 72 h" should be "72 hours".

72 hours after inoculation,

Line 432: "This results" should be "These results".

Line 434: "proving" should be "proved".

proved more effective

Line 435: Missing space after the period.

strains. Therefore,

Line 436: Awkward phrasing "so that strawberry could rapidly respond".

enabling strawberries to respond rapidly to F. solani infection and improve their disease resistance.

Line 443: Missing space after the period.

9). The

Line 463: Missing space after the period.

Line 467: Missing space between "widely" and "used".

Line 469: Repetition of "that".

Line 479: Missing space after the period.

scarce. This study

Line 490: "Inducing" should not be capitalized.

Line 491: Incorrect verb form "is" instead of "are".

Antibiotics are a pivotal

Line 508: Incorrect verb form "involves" instead of "involve".

mechanisms primarily involve

Line 533: "treated" should be "treatment".

36 hours of treatment with strains

Line 538: "induce" should be "inducing".

 inducing systemic resistance

Line 542: "on induce" should be "on inducing".

on inducing systemic resistance

Author Response

Dear Reviewer:

Thank you very much for your comments and professional advice.These opinions help to improve academic rigor of our article. Based on your suggestion, we had checked carefully all the some spelling-errors, grammatical errors and sentences, etc. in our manuscript. The changes in revised manuscript were highlighted with red. We hope that our work can be improved again. 

Meanwhile, we have added “the strains CMS5 and CMR12 showed strong effect in inhibiting mycelial growth of A. alternata” in discuss section, and the changes in revised manuscript were highlighted with red.

Reviewer 3 Report

Major comments

The presented research is relevant since the fight against root rot of strawberries is a complex task. To effectively protect crops from diseases, it is important to develop new effective and safe methods. This study showed that the isolated strains of Bacillus amyloliquefaciens were effective in controlling root rot of strawberries as they were able to suppress the development of fusarium and promote plant growth. They also had a significant effect on seedling growth, suggesting that they could be used as a biological control of root rot caused by Fusarium solani.

The article presents the results well, the chapter on methods needs to be improved and the review justifies the importance of developing biological control against Fusarium solani.

1. In the review, add information on the spread of the pathogen Fusarium solani to strawberries both in China and in the world as a whole. Justify why it is important to develop biocontrol against this pathogen.

2. In the materials and methods chapter, it is advisable to indicate, if it is known from which variety the Fusarium solani samples were isolated. Provide a link to the isolation of fungal cultures.

3. A description is given of the method for isolating bacteria from rhizosphere soil; these are generally accepted methods; a link must be provided. Also in clause 2.3 Screening of antagonistic strains you need to make a reference to the methodology, and in clause 2.6.

4. How was the species identity of Bacillus amyloliquefaciens confirmed?

5. Give an explanation of what kind of NA medium was used to isolate bacteria, provide a link.

6. It is necessary to explain by what method the ability of bacteria to fix nitrogen and secrete iron carriers was determined.

7. The crops used were Neopestalotiopsis clavispora, F. oxysporum and Alternaria alternata isolated in which year and from which crop?

8. The title of the work focuses on Bacillus amyloliquefaciens. It is necessary to explain the purpose for which the results of the effectiveness of other potential biological agents Neopestalotiopsis clavispora, F. oxysporum and Alternaria alternata against F. Solani are presented.

Author Response

Dear Reviewer:

Thank you very much for your comments and professional advice.These opinions help to improve academic rigor of our article. Based on your suggestion and request, we have made revised modifications on the revised manuscript.We hope that our work can be improved again. Furthermore, we would like to show the details as follows:

Question 1: In the review, add information on the spread of the pathogen Fusarium solani to strawberries both in China and in the world as a whole. Justify why it is important to develop biocontrol against this pathogen.

Answer: Based on your comments, we have added the information on the spread of the pathogen Fusarium solani to strawberries both in China and in the world in  Introduction section, and added the related references. The changes were marked in red in the revised manuscript.

Question 2: In the materials and methods chapter, it is advisable to indicate, if it is known from which variety the Fusarium solani samples were isolated. Provide a link to the isolation of fungal cultures.

Answer: Based on your comments, we have added the information of the pathogen Fusarium solani isolated from strawberries variety and isolation of fungal cultures in 2.1. Plant Pathogen and Plant Materials section, and added the related references. The changes were marked in red in the revised manuscript. 

Question 3: A description is given of the method for isolating bacteria from rhizosphere soil; these are generally accepted methods; a link must be provided. Also in clause 2.3 Screening of antago nistic strains you need to make a reference to the methodology, and in clause 2.6.

Answer: Based on your comments, we have added the related references about the method for isolating bacteria from rhizosphere soil, and we have added the related references in clause 2.3 and 2.6 section. The changes in revised manuscript were highlighted with red.

Question 4: How was the species identity of Bacillus amyloliquefaciens confirmed?

Answer: In 2.4. Identification of strains CMS5 and CMR12 section, we described the identification method of Bacillus amyloliquefaciens in detail. Strains CMS5 and CMR12 were identified preliminarily by gram staining and morphological characteristics and were further identified on the basis of 16S rDNA and gyrB sequences analysis.

Question 5: Give an explanation of what kind of NA medium was used to isolate bacteria, provide a link.

Answer: In 2.4. Identification of strains CMS5 and CMR12 section, we added the NA medium components. The changes in revised manuscript were highlighted with red.

Question 6: It is necessary to explain by what method the ability of bacteria to fix nitrogen and secrete iron carriers was determined.

Answer: In 2.7. Determination of plant growth-promoting traits of strains CMS5 and CMR12 section, we added the determined method of bacteria to fix nitrogen and secrete iron carriers, and added the related references. The changes in revised manuscript were highlighted with red.

Question 7: The crops used were Neopestalotiopsis clavispora, F. oxysporum and Alternaria alternata isolated in which year and from which crop?

Answer:  In“2.1. Plant Pathogen and Plant Materials”section, we added the crops used were Neopestalotiopsis clavispora in which year and F. oxysporum and Alternaria alternata isolated from which crop.The changes in revised manuscript were highlighted with red.

Question 8:The title of the work focuses on Bacillus amyloliquefaciens. It is necessary to explain the purpose for which the results of the effectiveness of other potential biological agents Neopestalotiopsis clavispora, F. oxysporum and Alternaria alternata against F. Solani are presented.

Answer:  In “4.1. Significance of exploring biocontrol resources for plant disease” section, we added some discussion content about the broad-spectrum activity of Bacillus amyloliquefaciens, and added the related references. The changes in revised manuscript were highlighted with red.
